# SENP6-Mediated deSUMOylation of VEGFR2 Enhances Its Cell Membrane Transport in Angiogenesis

**DOI:** 10.3390/ijms24032544

**Published:** 2023-01-29

**Authors:** Qi He, Zhenfeng Chen, Jieyu Li, Jinlian Liu, Zirui Zuo, Bingqi Lin, Ke Song, Chuyu Zhou, Haipeng Lai, Qiaobing Huang, Xiaohua Guo

**Affiliations:** Department of Pathophysiology, Guangdong Provincial Key Laboratory of Cardiac Function and Microcirculation, School of Basic Medical Sciences, Southern Medical University, Guangzhou 510515, China

**Keywords:** advanced glycation end products (AGEs), VEGFR2, SUMOylation, SENP6, angiogenesis

## Abstract

Angiogenesis is a significant pathogenic characteristic of diabetic microangiopathy. Advanced glycation end products (AGEs) are considerably elevated in diabetic tissues and can affect vascular endothelial cell shape and function. Regulation of the vascular endothelial growth factor (VEGF)-VEGF receptor 2 (VEGFR2) signaling pathway is a critical mechanism in the regulation of angiogenesis, and VEGFR2 activity can be modified by post-translational changes. However, little research has been conducted on the control of small ubiquitin-related modifier (SUMO)-mediated VEGFR2 alterations. The current study investigated this using human umbilical vein endothelial cells (HUVECs) in conjunction with immunoblotting and immunofluorescence. AGEs increased Nrf2 translocation to the nucleus and promoted VEGFR2 expression. They also increased the expression of sentrin/SUMO-specific protease 6 (SENP6), which de-SUMOylated VEGFR2, and immunofluorescence indicated a reduction in VEGFR2 accumulation in the Golgi and increased VEGFR2 transport from the Golgi to the cell membrane surface via the coatomer protein complex subunit beta 2. VEGFR2 on the cell membrane was linked to VEGF generated by pericytes, triggering the VEGF signaling cascade. In conclusion, this study demonstrates that SENP6 regulates VEGFR2 trafficking from the Golgi to the endothelial cell surface. The SENP6-VEGFR2 pathway plays a critical role in pathological angiogenesis.

## 1. Introduction

Chronic diseases such as diabetes mellitus adversely affect the diffusive microvascular system [1]. Diabetic microangiopathy is the most common complication of diabetes and is one of the main symptoms that endanger the lives of diabetics. It can affect several organs, including those with diabetes nephropathy, diabetic retinopathy, and diabetic peripheral neuropathy [2]. Angiogenesis is the primary pathogenic hallmark of diabetic proliferative retinopathy, which predominantly affects the structure and function of the retinal microvascular system [3,4,5].

Angiogenesis is the process of generating and maintaining arteries, involving many cells and substances [6]. It is a complex physiological process that is strictly controlled by specific biomolecules produced in vivo and develops a new capillary network from the outer membrane of existing blood vessels, thus forming a complete blood supply system. A mature, complete, complex, and ordered vascular venous system is vital for maintaining homeostasis [7]. Early in the process of angiogenesis, there is an increase in vascular permeability [8]. Physiologic angiogenesis and pathological angiogenesis are the two fundamental types of vascular angiogenesis. Aberrant and uncontrolled angiogenesis triggers an inflammatory reaction in the body and encourages the emergence and progression of several microangiopathies [9,10].

Amadori products can be created by a spontaneous reaction between glucose or other reduced monosaccharides and the reducing amino group found at the end of proteins and nucleic acids [11]. After further rearrangement, dehydration, condensation, and other processes, these advanced glycation end products (AGEs) constitute irreversibly cross-linked polyphase fluorescent derivatives [12]. There is a link between AGEs and the vascular complications associated with diabetes mellitus. The synthesis of AGEs is markedly accelerated by hyperglycemia or the maintenance of hyperglycemia. This leads to a buildup of AGEs in the body, the creation of AGEs more rapidly than they are cleared, and ultimately damaging cells and tissues [13]. The oxidative stress response, which destroys proteins and increases endothelial cell dysfunction, is reportedly triggered by interaction between AGEs and the receptor for advanced glycation end products [14].

Vascular endothelial growth factors (VEGFs) and VEGF receptors (VEGFRs) are necessary during development and in adult tissues. They maintain and modify existing blood vessels while balancing the formation of new ones [15]. There is now a consensus that VEGFR2 is the main VEGFR that mediates the angiogenic effects of VEGF. VEGFR2 is the key signaling VEGFR in blood vascular endothelial cells [16]. VEGF signaling intensity is regulated by VEGFR2 cell surface concentration [17,18]. The VEGF signaling pathway is crucial in the development of diabetic retinopathy [4]. A variety of downstream signals, including ERK, PLC/MARCKS, and p38 are activated when VEGF binds to the receptor VEGFR2. These signaling pathways influence the growth, migration, and permeability of endothelial cell barriers [19].

Small ubiquitin-related modifier (SUMO) protein post-translational modification is now acknowledged as one of the most significant regulatory protein modifications in eukaryotic cells [20]. Sentrin/SUMO-specific proteases (SENPs) may readily reverse the dynamic process of SUMOylation, which is mediated by the enzymes E1, E2, and E3 in living things [21]. SUMOylation and SENP-mediated deSUMOylation are both extremely dynamic processes throughout the cell cycle [22]. The human SENP family consists of six members, SENP1, SENP2, SENP3, SENP5, SENP6, and SENP7 [23]. SENPs are associated with immune infiltration, as well as some transcription factors that control apoptosis, proliferation, the cell cycle, carcinogenesis, and invasion [24,25,26,27,28].

In the present study, we show that following stimulation of endothelial cells by AGEs, SENP6 de-SUMOylated VEGFR2, VEGFR2 accumulation in the Golgi was reduced and VEGFR2 transfer from the Golgi to the cell membrane surface via the coat protein complex subunit β2 (COPB2) was increased. VEGFR2 on the cell membrane is linked to VEGF produced by pericytes, triggering a VEGF signaling cascade. In conclusion, this study aimed to investigate the effect of AGEs on angiogenesis in HUVECs and the mechanism of SENP6-mediated VEGFR2 deSUMOylation in angiogenesis.

## 2. Results

### 2.1. AGEs Upregulate VEGFR2 Expression by Inducing NF-E2-Related Factor-2 Translocation into the Nucleus

The transcription factor NF-E2-related factor-2 (Nrf2) can be activated during vascular development, leading to its segregation from the repressor Keap1 and its relocation from the cytoplasm to the nucleus, where it is implicated in angiogenesis regulation [29,30]. The effect of AGEs on Nrf2 was investigated. We used Kit-8 (CCK8) experiments to verify the cytotoxicity of AGEs on HUVECs. The results showed no cytotoxicity to HUVECs after AGEs (100 µg/mL) stimulation (Appendix A). AGEs (100 µg/mL) were added to HUVECs to stimulate them for 0, 2, 4, and 6 h. Western blot was conducted to identify Nrf2 expression in the nucleus and plasma after cytoplasmic and nuclear proteins had been extracted (Figure 1A). With increasing stimulation time, Nrf2 expression decreased in the cytoplasm and increased in the nucleus, and there was a statistically significant difference between AGEs stimulation times of 2 h and 0 h (Figure 1B,C). This indicates that AGEs induce Nrf2 translocation into the nucleus in a time-dependent manner.

ML385 reportedly prevents Nrf2 from moving from the cytoplasm into the nucleus. Next, we investigated how ML385 affected Nrf2 in HUVECs. HUVECs were stimulated with AGEs for 6 h after being pretreated with ML385 (5 mM) for 1 h. Immunofluorescence data indicated that as AGEs were administered, Nrf2 in the nucleus increased greatly compared to the control group. When AGEs were combined with ML385 (ML385 + AGEs group), Nrf2 in the nucleus decreased significantly. This indicates that ML385 inhibits the translocation of Nrf2 into the nucleus in HUVECs (Figure 1D,E). After AGE stimulation and pretreatment with the inhibitor ML385, VEGFR2 expression in HUVECs was detected via immunoblotting. There were statistically significant differences between VEGFR2 expression levels in the AGEs group and the control group and between the ML385 + AGEs group and the AGEs group. This indicates that the increase in VEGFR2 expression induced by AGEs is attenuated by inhibiting Nrf2 nucleation (Figure 1F,G). Pre-transfection with Nrf2 siRNA and NC siRNA independently indicated that Nrf2 siRNA downregulated Nrf2 expression (Appendix A). AGEs were then administered to activate VEGFR2 expression levels, which were detected via immunoblotting. This yielded the same outcomes as addition of the inhibitor ML385 (Figure 1H,I). This suggests that decreasing Nrf2 levels may alleviate the overexpression of VEGFR2 caused by AGEs. These findings imply that AGEs increase VEGFR2 expression by causing Nrf2 translocation into the nucleus.

### 2.2. Inhibiting Nrf2 Entry Decreases AGE−Induced Angiogenesis

In transwell assays, the number of migrating cells rose considerably in the AGEs group compared to the control group, whereas the number of migrating endothelial cells was reduced in the ML385 + AGEs group compared to the AGEs group (Figure 2A,B). In cell scoring assays the migrating area of cells in the AGEs group rose substantially compared to the control group, but the migrating area of endothelial cells in the ML385 + AGEs group reduced dramatically compared to the AGEs group (Figure 2F,G). Matrigel test results indicated that the total length of tubules and the number of branching nodes were significantly increased in the AGEs group compared to the control group. In the ML385 + AGEs group, the number of branching nodes and the total length of vessels were significantly reduced compared to the AGEs group, suggesting that the reduction in Nrf2 entry reduced the tubular contribution of AGE−induced endothelial cells. All differences were statistically significant (Figure 2C–E). This demonstrates that blocking Nrf2 entry reduces AGE−induced angiogenesis.

### 2.3. SENP6−Reduced HUVECs Attenuate AGE−Induced Angiogenic Signaling

Transcriptomic sequencing of AGE−treated HUVECs revealed that SENP6 was the most substantially elevated member of the SENP family (Figure 3A). SENP6 expression was subsequently measured via qPCR and immunoblotting, both of which revealed a significant increase 8 h after AGE administration (Figure 3B–D). In vitro models of HUVECs migration and tube formation were then used to assess whether SENP6 mediated angiogenic signaling.

In transwell assays, the number of migrating cells rose considerably in the NC siRNA + AGEs group compared to the NC siRNA group, whereas the number of migrating endothelial cells decreased in the SENP6 siRNA + AGEs group compared to the NC siRNA + AGEs group (Figure 3E,F). In scratch assays, the migrating area of cells in the NC siRNA + AGEs group increased significantly compared to the NC siRNA group, whereas the migrating area of endothelial cells in the SENP6 siRNA + AGEs group decreased significantly compared to the NC siRNA + AGEs group (Figure 3J,K). The overall length of tubules and the number of branching nodes were improved considerably in the NC siRNA + AGEs group compared to the NC siRNA group in the Matrigel test. The number of branching nodes and overall length of arteries were considerably decreased in the SENP6 siRNA + AGEs group compared to the NC siRNA + AGEs group (Figure 3G–I). These findings indicate SENP6′s crucial function in modulating AGE−induced angiogenic signaling.

### 2.4. VEGFR2 SUMOylation Retains VEGFR2 in the Golgi

SUMOylation−modified VEGFR2 reportedly accumulates at the Golgi in SENP1-deficient endothelial cells. In the present study, an immunoprecipitation technique was used to measure amounts of VEGFR2 SUMOylation. The level of modification decreased after stimulation with AGEs (Figure 4C). The endopeptidase SENP6 deconjugates SUMO from substrate proteins. We surmised that the equilibrium of VEGFR2 SUMOylation was dysregulated because SENP6 expression was elevated. Immunofluorescence techniques indicated that VEGFR2 accumulates in the Golgi under unstimulated conditions, and colocalizes with GM130. When HUVECs were stimulated with AGEs; however, VEGFR2 accumulation in the Golgi and colocalization with GM130 were decreased (Figure 4D–F). Similarly, after transfection with SENP6 siRNA, there was a significant increase in VEGFR2 in the Golgi (Figure 4G). These findings imply that HUVECs stimulated by AGEs undergo a reduction in VEGFR2 SUMOylation modification, causing a reduction in accumulation in the Golgi apparatus and a reduction in colocalization with Golgi markers.

### 2.5. The Vesicular Transport Protein COPB2 Mediates Enhanced Transport of VEGFR2 to Membranes

Endothelial cells were stimulated with AGEs to analyze VEGFR2 expression. After 24 h there was considerable variation in total VEGFR2 over time (Figure 5A,B). A kit was used to separate and extract proteins from the cytoplasm and cell membrane to investigate VEGFR2 expression. VEGFR2 in the cytoplasm was greatly increased at 24 h (Figure 5C,D), whereas VEGFR2 in the cell membrane started to increase after 16 h (Figure 5E,F). This implies that endothelial cells were activated by AGEs and that VEGFR2 was translocated to the cell membrane. In transcriptome analyses, after AGE stimulation, differential gene ontology (GO) was enriched, including upregulation of many genes associated with Golgi vesicle transport. This indicated increased vesicle transport activity from the Golgi to the cell membrane (Figure 5G). qPCR was then used to validate the top-ranked genes, and COPB2 was considerably upregulated (Figure 5H). COPB2, a transporter protein, has been shown to play a key role in embryonic development and tumor progression. It has also been associated with cell proliferation, survival, invasion, and metastasis [31,32,33]. Levels of COPB2 protein were then determined via immunoblotting, and they had also significantly risen after AGE stimulation (Figure 5I,J). The same result was evident in endothelial cells after AGE stimulation, in which there was a considerable increase in COPB2 release as determined via immunofluorescence (Figure 5K). COPB2 expression was silenced using siRNA ((Appendix A), and when cell membrane VEGFR2 was extracted to assess its level of expression, it had been markedly suppressed (Figure 5L,M). Intracellular VEGFR2 translocation to the cell membrane via the vesicular transport protein COPB2 may be enhanced when AGEs stimulate endothelial cells.

### 2.6. AGEs Stimulate Increased Secretion of VEGF by Pericytes and Promote Endothelial Cell Angiogenesis

Pericytes were treated with different types of conditioned media to investigate the effects on VEGF expression in endothelial cells. Endothelial cells were stimulated with AGEs, VEGF expression was assessed, and there were no significant differences over time (Figure 6A,B). There was an increase in total VEGF expression in pericytes, with a significant difference at 8 h (Figure 6C,D). Enzyme-linked immunosorbent assays of VEGF expression in culture medium supernatants indicated an increase over time, with a significant difference at 16 h (Figure 6E,F). HUVECs were stimulated with different types of pericyte-conditioned media (PCM), and migration, wound healing, and tube formation assays were performed. In cell migration assays there was a significant increase in the number of migrating cells in the PCM+AGEs group compared to the PCM alone group (Figure 6J,K), and germination and tubule formation were significantly increased (Figure 6G,I). In scratch assays, the migrating area of cells in the PCM+AGEs group increased significantly (Figure 6L,M). This indicates that AGEs induce VEGF secretion by pericytes and promote endothelial cell angiogenesis.

## 3. Discussion

Angiogenesis is a multifaceted process that is crucial for growth and development, organ and tissue regeneration, and a variety of pathological diseases [34]. A balance between related promoters and inhibitors controls angiogenesis, keeping it within the range of normal physiological limitations [7].

For many years, scientists have been studying the effects of AGEs on endothelial cells. However, whether AGEs promote or inhibit HUVECs proliferation and migration is still being debated. Our previous study showed that AGEs (100 µg/mL) promoted the proliferation, migration, and tube formation of HUVECs [35,36]. Accordingly, several experimental studies have come to the same conclusions as ours [37,38,39]. However, there are also kinds of literature showing that AGEs inhibit migration and tubule formation [40,41]. One literature systematically addresses this paradox effect of glycosylation on angiogenesis, as AGEs can play contradictory roles at different stages of angiogenesis [42]. Possible explanations include (1) different AGE concentrations in different studies and (2) different AGE preparation methods in different studies. (3) Different culture conditions may result in the production of different active compounds in AGEs.

In the present study stimulating HUVECs with AGEs resulted in a considerable increase in VEGFR2 expression, followed by a decrease in VEGFR2 expression after inhibition of Nrf2 entry or downregulation of Nrf2 expression. This suggests that VEGFR2 may function as a downstream gene of Nrf2. Nrf2 is a transcriptional regulator that regulates the expression and coordinated activation of many defense genes that encode detoxification enzymes and antioxidant proteins [43]. It is evidently involved in the control of angiogenesis and is activated and translocated from the cytoplasm into the nucleus [44,45]. AGEs stimulation promoted Nrf2 translocation into the nucleus, and ML385 prevented AGE−induced Nrf2 translocation into the nucleus. Downregulation of Nrf2 expression and inhibition of Nrf2 nucleation both reduced VEGFR2 expression in HUVECs, and suppression of Nrf2 nucleation also reduced endothelial cell vascularization. Many groups have investigated VEGFR2 phosphorylation, acetylation, ubiquitination, and other post-translational changes, but its SUMOylation has received fewer attention [46]. SUMOylation is reversible and is achieved by SUMO protease, which removes SUMO cells from the substrate [47]. The SUMO protease SENP6 is essential for the proliferation and survival of almost all known tumor cell lines [48]. In the present study, we found that SENP6 expression increased after AGEs stimulation of HUVECs, but the exact mechanism of the increase deserves further investigation. Furthermore, we discovered that decreasing SENP6 expression significantly inhibited AGE−induced angiogenesis.

In the current study, AGEs reduced VEGFR2 SUMOylation modifications. It has been suggested that hyperglycemia may induce the accumulation of VEGFR2 in the Golgi apparatus, and reduce its surface expression [49]. Similarly, immunofluorescence after AGEs activation of HUVECs revealed a decrease in VEGFR2 colocalization with Golgi markers, indicating a decrease in VEGFR2 accumulation inside the Golgi apparatus. VEGFR2 is translocated to the membrane where it functions as a membrane protein. The process of VEGFR2 transport to the membrane is critical for HUVEC germination. A constant supply of VEGFR2 is required for vascular growth, and translocation of newly synthesized VEGFR2 to the plasma membrane is a key determinant of angiogenesis [50]. KIF13B is reportedly the key molecular motor necessary for the transport of VEGFR2 from the Golgi apparatus, and its distribution to the endothelial cell surface promotes angiogenesis [51]. In the present study, there was increased cytosolic VEGFR2 expression in endothelial cells after AGE stimulation. Gene ontology enrichment was indicative of upregulation of the Golgi vesicle transport-associated protein COPB2, which was confirmed at the mRNA and protein levels by detecting a decrease in cytosolic VEGFR2 expression when COPB2 expression was downregulated. Even though COPB2 may assist VEGFR2 translocation from the cytoplasm to the cell membrane, the specific mechanism by which COPB2 acts requires additional investigation.

VEGF is currently the most studied pro-angiogenic factor [52]. The signaling pathways activated by VEGF play an irreplaceable role in the entire process of angiogenesis, from the increase in vascular permeability, basement membrane breakdown, endothelial cell proliferation, and differentiation to final tube formation, and play an important pathophysiological role in angiogenesis, embryonic development, wound healing, collateral circulation, inflammatory response, and tumor development [53,54,55]. VEGF can act on endothelial cells in an autocrine and paracrine manner, thereby promoting angiogenesis [56,57,58]. Pericytes are outer membrane cells with multiple cytoplasmic protrusions located near the basement membrane of capillaries and small post-capillary veins. Angiogenesis requires new tissue-specific vascular cells, namely the associated network of supporting stromal cells (pericytes) and extracellular matrix (ECM), to modulate the effects of vascular endothelial growth factor (VEGF) and basic fibroblast growth factor (BFGF)-mediated angiogenesis [59,60,61]. Our experimental results also confirm that AGEs induce endothelial angiogenesis with VEGF that is not derived from source endothelial cells, but from pericytes, which in turn act on endothelial cells.

In conclusion, our findings show that AGEs can prompt angiogenesis by increasing SENP6 expression, which de-SUMOlates VEGFR2 and increases its transport to the cell membrane. These findings contribute to a better understanding of the role of AGEs and their signaling pathways in the development of diabetic microvascular complications, and SENP6 may be a candidate target for treating the disease.

## 4. Materials and Methods

### 4.1. Culture of HUVECs

HUVECs (Sciencell, Carlsbad, CA, USA) were cultured in ECM (Sciencell, Carlsbad, CA, USA) supplemented with FBS (10%). Cells were cultured at 37 °C in a humidified incubator with 5% CO_2_ atmosphere. When about 90% confluence degree is reached, cells are cultured in serum-free medium for 12 h, and then used for the following experiment.

### 4.2. Isolation of Retinal Microvascular Pericytes

As previously described [62], primary retinal microvascular pericytes (RMPs) were isolated from the retinal microvessels of 3-week-old male weanling rats. Fresh rat retinas were isolated and minced into homogeneous fragments in precooled PBS buffer before being suspended and incubated in 0.2% type I collagenase for 20 min at 37 °C. DMEM (GBCBIO Technologies, Guangzhou, China) containing low glucose (5 mmol/L) and 20% FBS (L-DMEM-20) was added, and the suspension was gently mixed and then filtered through 55 m filters. After collecting the final filtrate, it was centrifuged at 500× *g* for 5 min at 4 °C. The pellets were resuspended in DMEM containing 20 mmol/L glucose and 20% FBS before being seeded in culture dishes. The dishes were rinsed after 72 h of incubation to remove loosely adherent cellular contamination, and the medium was replaced with L-DMEM-20 on days 3–5. Cells were digested with trypsin when they reached 80–90 percent confluence, and digestion was stopped after 1–2 min when contaminating cells began to detach. By gently swirling the dish, removing the detached cells, discarding the medium, and adding new trypsin to the cells for passage, the detached cells were removed. Three-week-old male weanling rats were provided by the Laboratory Animal Centre of Southern Medical University (Guangzhou, China). All animals were maintained in a specific condition (temperature 23–25 °C; humidity 50 ± 5%) with a 12 h light/12 h dark cycle and fed standard food and water. All experimental procedures were approved by the Institutional Animal Care and Use Committee of Southern Medical University.

### 4.3. Preparation of AGEs-Modified Bovine Serum Protein

AGEs were prepared as previously reported [63], and all stages of preparation should be carried out under aseptic circumstances. Aseptic conditions should be used throughout the entire preparation process. In summary, d-glucose (250 mmol/L) and BSA (GBCBIO Technologies, Guangzhou, China) (150 mmol/L, pH 7.4) were incubated for 8 weeks at 37 °C in PBS. After that, the fluid was transferred to a 50 mL ultrafiltration column and centrifuged for 45 min at 4 °C at 4000 rpm. The liquid in the ultrafiltration column is then filtered, divided into Ep tubes, and stored in the storage tank at −80 °C.

### 4.4. RNA Extraction and Real-Time Quantitative Polymerase Chain Reaction

Total RNA was isolated from cells using Trizol (GBCBIO Technologies, Guangzhou, China) reagent according to the instruction of the manufacturer. RNA quality and concentration were tested by using a BioDrop spectrophotometer. RNA was converted to cDNA using the RT-PCR Mix for qPCR (Dongsheng Biotech, Guangzhou, China) following the manufacturer’s instructions. Briefly, qRT-PCR was performed in a 7500 Fast Real-Time PCR System (Applied Biosystems 7500, Waltham, MA, USA) using Power Green qPCR Mix (Dongsheng Biotech, Guangzhou, China) to quantify mRNA expression levels. The primers used for qRT-PCR analysis are listed in Table 1.

### 4.5. Extraction of Cytoplasmic Protein and Nuclear Protein

When using the nucleoprotein extraction kit, adhere to the directions (Bestbio, Shanghai, China). Cells were centrifuged at 4 °C and 12,000 rpm for 10 min after being washed twice or three times with precooled PBS and the lysate was prepared in advance. Take the cytoplasmic protein-containing supernatant and place it in a new tube. The remaining precipitates were then centrifuged for 10 min, separated from the supernatant, and once washed with precooled PBS. Add lysate to the precipitation, then shake it rapidly with a high-speed vortex for 15 s. After it has been on ice for 40 min, shake it with a high-speed vortex for 15 s every 4 min. Then spin at 4 °C and 1200 rpm for 10 min. Finally, add the supernatant to a fresh nucleoprotein tube.

### 4.6. Extraction of Membrane Protein and Cytoplasmic Protein

Operate according to the instructions of membrane protein and cytoplasmic protein extraction kit (Bestbio, Shanghai, China). Take 5 × 10^6^ cells, incubate them at 4 °C for 2–3 min at 500 g, and then wash the cells twice with cold PBS to make the extract. Shake between 2 and 8 °C for 30 to 1 h, 12,000 rpm at 4 °C. Take a 10 min water bath at 37 °C and centrifuge for three minutes after inhaling the supernatant into another precooled centrifuge tube for five minutes. The upper layer is a cytoplasmic protein, and the lower layer is a membrane protein. A cytoplasmic protein makes up the upper layer, while a membrane protein makes up the lower layer. The aforementioned protein extracts are weighed out and sub packaged in a −80 °C freezer for later usage or storage.

### 4.7. Western Blot Analysis

Using the RIPA lysis buffer (GBCBIO Technologies, Guangzhou, China), proteins were extracted from HUVECs or pericytes and then separated by 4–10% SDS-PAGE before being transferred to the PVDF membrane (Millipore, Burlington, MA, USA). The blot was incubated in 5% bovine serum albumin (BSA) (GBCBIO Technologies, Guangzhou, China) for at least 1 h and then reacted with primary antibodies at 4 °C overnight. Next, washing for 15 min in TBST (GBCBIO Technologies, Guangzhou, China) three times, the membranes were incubated with the secondary antibodies for 1 h, and the signal was detected by chemiluminescence. Finally, the result of band density was carried out by the Tanon imaging station (Tanon, Shanghai, China) and analyzed with Image J. Primary antibodies used for Western blotting were β-actin (1:10,000, Z0529, Ray antibody Biotech, Beijing, China), VEGFR2 (1:1000, 26415-1-AP, Proteintech, Rosemont, IL, USA), VEGF-A (1:1000, YT5108, ImmunoWay Biotechnology, Plano, TX, USA), SENP6 (1:1000, AF0277, Affinity Biosciences, Beijing, China), SUMO-2/3 (1:1000, YC0075, ImmunoWay Biotechnology, Plano, TX, USA), COPB2 Rabbit pAb (1:1000, A7036, Abclonal, Woburn, MA, USA), Nrf2 (1:1000, YT3189, ImmunoWay Biotechnology, Plano, TX, USA), Lamin B1 (1:1000, YT2522, ImmunoWay Biotechnology, Plano, TX, USA), respectively. Goat anti-mouse IgG-HRP (1:10,000, 3001, Ray antibody Biotech, Beijing, China) and Goat anti-Rabbit IgG-HRP (1:10,000, 3002, Ray antibody Biotech, Beijing, China) were used as secondary antibodies.

### 4.8. Co-Immunoprecipitation

For immunoprecipitation, the HUVECs cracking liquid was extracted using 200 L cell lysis buffer (Beyotime Biotechnology, Shanghai, China) (for the identification of SUMOylated proteins, 1 mM N-ethylmaleimide (deSUMOylase inhibitor) was added to cell lysates). In brief, as previously described [49], 100 L of total protein and 1 L of VEGFR2 antibody were incubated at 4 °C for an overnight period before being combined for two hours with 20 L of protein A protein/G mixed magnetic beads. Then three to five times with PBS. The immunological complexes were subjected to SDS-PAGE and then an immunoblot using an antibody specific for the second protein (SUMO2/3, 1:1000, YC0075, ImmunoWay Biotechnology, Plano, TX, USA).

### 4.9. Enzyme-Linked Immunosorbent Assay (ELISA)

The pericyte medium supernatant was collected, and the biotinylated antibody (VEGF, E-EL-R2603c, Elabscience Biotechnology, Wuhan, China) was added as described in the literature [64], the enzyme conjugate working solution and the color developer were added separately, the reaction was terminated, and the optical density OD of each well was immediately measured at 450 nm using a multi-well high-performance reader. The absorbance values of known concentrations of the standards were used to create a standard curve, and the corresponding sample concentrations could be calculated using the measured absorbance values.

### 4.10. Immunofluorescent Test

Cells were fixed with 4% paraformaldehyde for 15 min, permeabilized with 0.1% Triton X-100 for 10 min, and blocked with goat serum for 60 min. Subsequently, the cells were incubated with respective primary antibodies overnight at 4 °C and stained with second antibodies for 60 min at room temperature. DAPI was used to stain the cell nucleus. The cells were washed with PBS buffer between each step. The stained cells were observed under confocal laser-scanning microscopy (LSM780, Carl Zeiss, Oberkochen, BW, Germany). The primary antibodies used for immunofluorescence were anti-VEGFR2 (1:200, sc-393163, Santa Cruz, CA, USA), COPB2 Rabbit pAb (1:200, A7036, Abclonal, Woburn, MA, USA), and anti-GM130 (1:200, A5344, Abclonal, Woburn, MA, USA), respectively.

### 4.11. Transfection of siRNA

The cell transfection was conducted following the instruction of the manufacturer. Cells were grown in a medium without antibiotics for 24 h with target siRNA. The siRNA and negative control siRNA were purchased from GenePharma (Shanghai, China). Two days after transfection, the cells were used for the following experiments. The siRNA sequences are presented in Table 2.

### 4.12. Cell Counting Kit-8 (CCK-8)

CCK-8 was used to measure cytotoxicity (Dojindo Molecular Technologies Inc., Kumamoto, Japan). Cells were seeded in 96-well culture plates and then treated in groups. After removing the media, CCK-8 (0.5 mg/mL) was added to each well for 4 h. At 450 nm, the absorbance was measured. HUVEC proliferation was assessed directly using optical density (OD).

### 4.13. Endothelial Cell Transwell Assay

Endothelial cells were placed in the Transwell culture dish’s upper chamber, and 800 µL of the basic medium was placed in the lower chamber. At 24 h incubation period, the cells were subjected to 4% paraformaldehyde for 10 min before going through two or three rounds of PBS washing. After the cells have been colored for ten minutes, carefully brush away any non-migrated cells with a cotton swab after washing them three to four times in PBS. Observe endothelial cells using an upright microscope (Axio Imager Z, Carl Zeiss NTS Ltd., Oberkochen, Germany), and capture three different visual fields from each chamber.

### 4.14. Scratch Assay for Migration Cells

A procedure that was previously reported was changed for this test [65]. Transfected cells were seeded into six-well plates, and exposed to 24 h of serum deprivation in a serum-free medium, and then a 200 μL pipette tip was used to artificially puncture the confluent cell monolayer. All media were switched to new media and continued to be cultured for 24 h after infection after 1 h. By assessing the variation in wound areas, cell migration was examined. A camera mounted on a Zeiss Axiovert microscope (Carl Zeiss NTS Ltd., Oberkochen, Germany)was used to take pictures.

### 4.15. EC Tube Formation Assay

Pour 50 L of Matrigel (Corning, NY, USA) evenly into each hole of 96-well plates, and then incubate the plates for 30 min at 37 °C. In a medium devoid of serum, early passage cells were reconstituted. For every hole l (2 × 10^4^) in the matrix liquid, add 100. After 30 min, add fresh medium, and then incubate the samples for 4–24 h at 37 °C with 4 holes for each condition group. To picture the tiny pipe network in each well from 3 to 5 randomly selected fields, an inverted microscope (IX71, Olympus, Tokyo, Japan) is employed. ImageJ is then used to evaluate the results.

### 4.16. Statistical Analysis

GraphPad Prism (version 8.0, San Diego, CA, USA) was used to evaluate the data. All data are presented as means ± SD of at least three distinct investigations. One-way ANOVA was used for statistical comparisons, and then a proper post hoc test was run. *p* < 0.05 was chosen as the significant threshold.

## Figures and Tables

**Figure 1 ijms-24-02544-f001:**
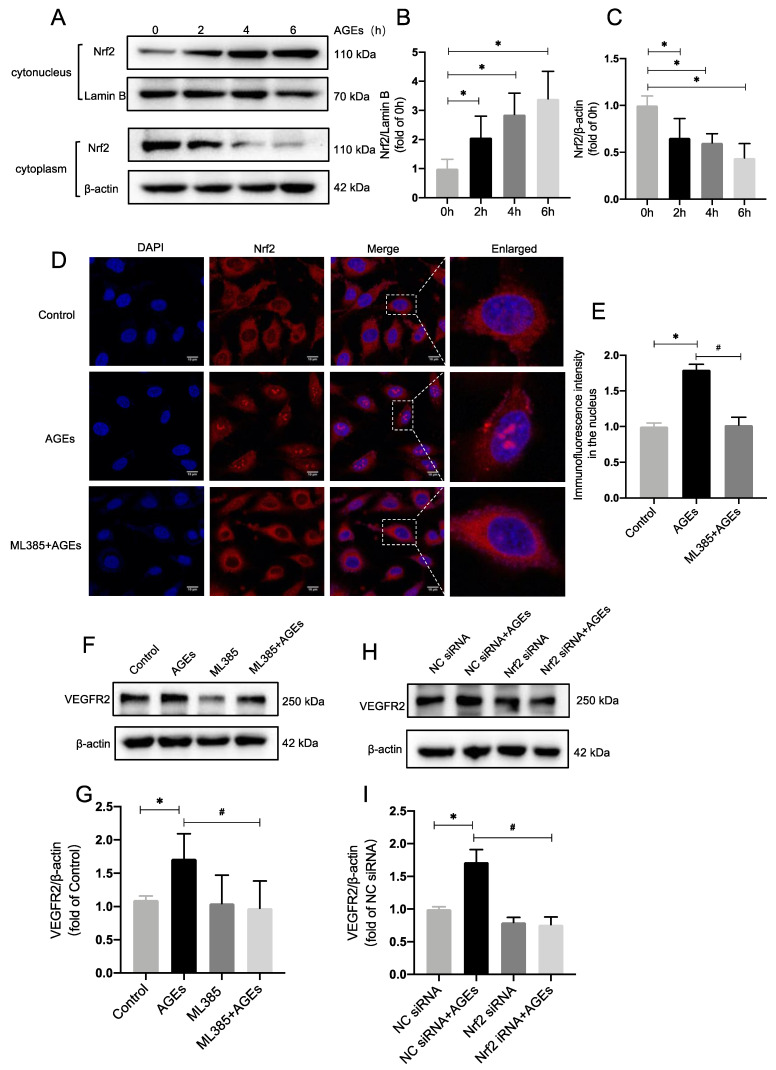
AGEs upregulate VEGFR2 expression by inducing Nrf2 translocation into the nucleus. (**A**–**C**) 100 µg/mL AGEs were applied to HUVECs to stimulate them for 0, 2, 4, and 6 h, *n* = 5, * *p* < 0.05 vs. 0 h. (**D**,**E**) Immunofluorescence staining, Nrf2 (red); nucleus (blue), and HUVECs were stimulated with AGEs for 6 h after being pretreated with ML385 (5 mM) for 1 h, *n* = 4, Scale bar: 10 μm. (**F**,**G**) AGEs stimulation and pretreatment with the inhibitor ML385, VEGFR2 expression in HUVECs was detected. *n* = 8. * *p* < 0.05 vs. control, ^#^
*p* < 0.05 vs. AGEs. (**H**,**I**) Pre-transfection of Nrf2 siRNA and NC siRNA, detect the expression of VEGFR2. *n* = 7, * *p* < 0.05 vs. NC siRNA, ^#^
*p* < 0.05 vs. NC siRNA + AGEs. * and ^#^ indicate significance between the indicated groups.

**Figure 2 ijms-24-02544-f002:**
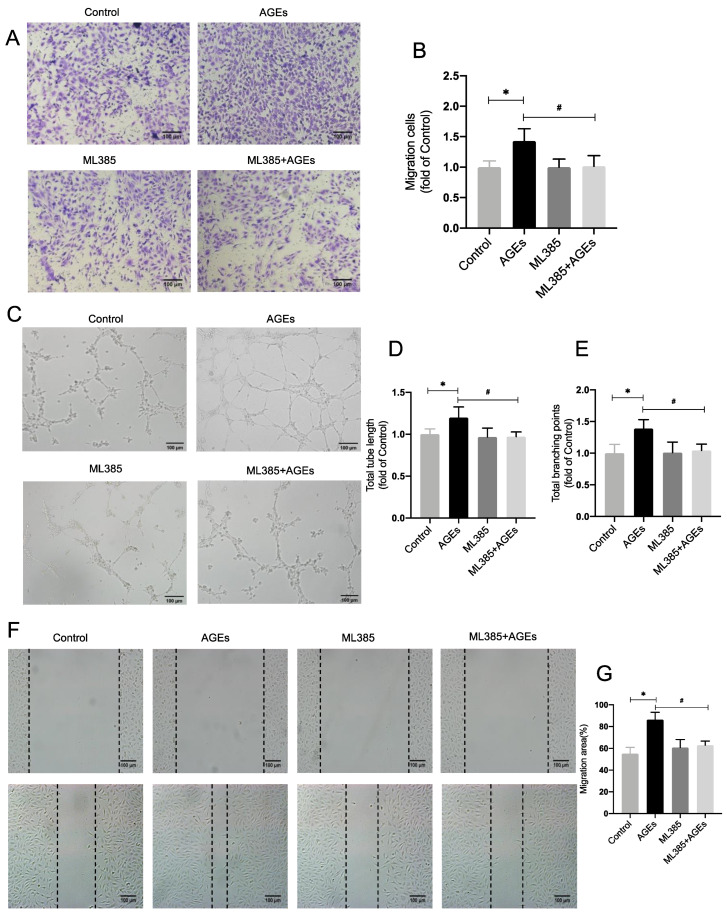
Inhibiting Nrf2 entry decreases AGE−induced angiogenesis. (**A**,**B**) Transwell migration assay, inhibition of Nrf2 nuclear translocation alleviated the endothelial migration caused by AGEs in HUVECs, *n* = 4, * *p* < 0.05 vs. control, ^#^
*p* < 0.05 vs. AGEs. Scale bar: 100 μm. (**C**–**E**) Inhibition of Nrf2 nuclear translocation alleviated the endothelial tube formation and lessened the total branching points caused by AGEs in HUVECs. *n* = 4, * *p* < 0.05 vs. control, ^#^
*p* < 0.05 vs. AGEs. Scale bar: 100 μm. (**F**,**G**) Scratch wound healing, *n* = 6, * *p* < 0.05 vs. control, ^#^
*p* < 0.05 vs. AGEs. Scale bar: 100 μm. * and ^#^ indicate significance between the indicated groups.

**Figure 3 ijms-24-02544-f003:**
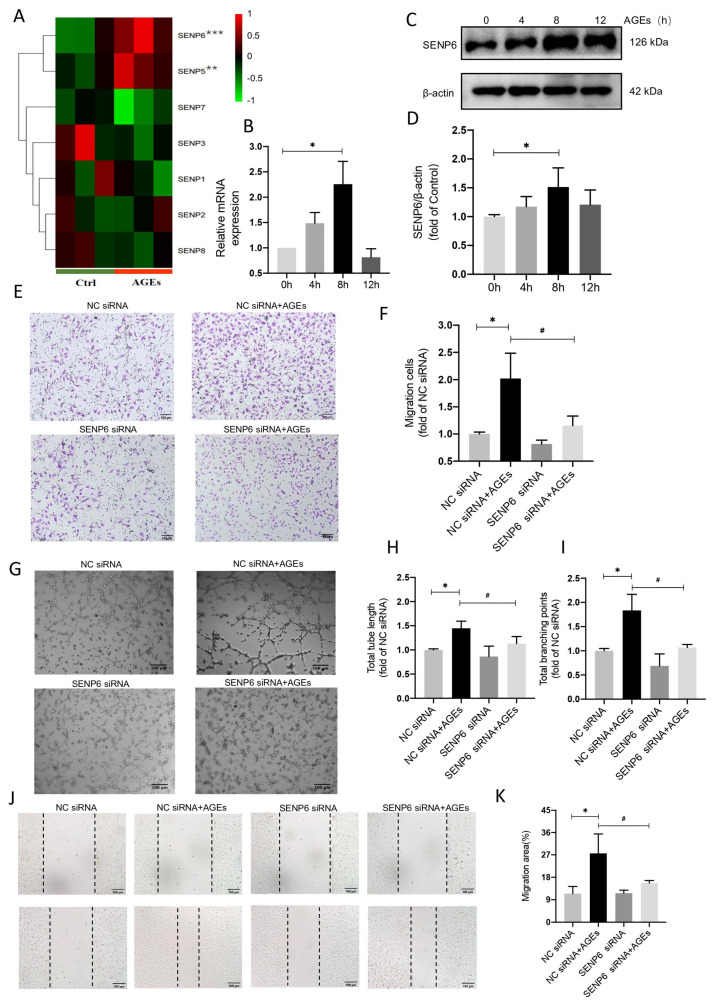
SENP6−reduced HUVECs attenuate AGE−induced angiogenic signaling. (**A**) Heat map of mRNA showing the differences in SENPs expression between the control and AGEs groups. “Red” indicates up-regulation and “Green” indicates doregulation. ** indicates that *p* < 0.01 and *** indicates that *p* < 0.001 between the indicated groups. (**B**) Detection of SENP6 mRNA expression by qPCR at 0, 4, 8, and 12 h of AGE−stimulated HUVECs, *n* = 4, * *p* < 0.05 vs. 0 h. (**C**) The temporal effect of AGE stimulation on HUVECs was investigated using Western blot, which showed SENP6 protein expression at 0, 4, 8, and 12 h. (**D**) Analysis of protein expression level results, *n* = 6, * *p* < 0.05 vs. 0 h. AGE−induced angiogenic responses were inhibited by SENP6 silencing (Appendix A). HUVECs were transfected with NC siRNA or SENP6 siRNA for 48 h. Cells were subjected to HUVECs migration and tube formation in response to AGEs. (**E**) Transwell migration assay for the migration ability. (**F**) Analysis of migration assay results, *n* = 3, * *p* < 0.05 vs. NC siRNA, ^#^
*p* < 0.05 vs. NC siRNA + AGEs, Scale bar: 100 μm. (**G**) HUVECs tube formation in a Matrigel assay. Representative images are shown in (**G**). (**H**,**I**) Analysis of total tube length and total branching points results, *n* = 4, * *p* < 0.05 vs. NC siRNA, ^#^
*p* < 0.05 vs. NC siRNA + AGEs, Scale bar: 100 μm (**J**) HUVECs migration by a scratch assay for indicated times. (**K**) Wound healing (% closure) was quantified, *n* = 5, * *p* < 0.05 vs. NC siRNA, ^#^
*p* < 0.05 vs. NC siRNA + AGEs, Scale bar: 100 μm. * and ^#^ indicate significance between the indicated groups.

**Figure 4 ijms-24-02544-f004:**
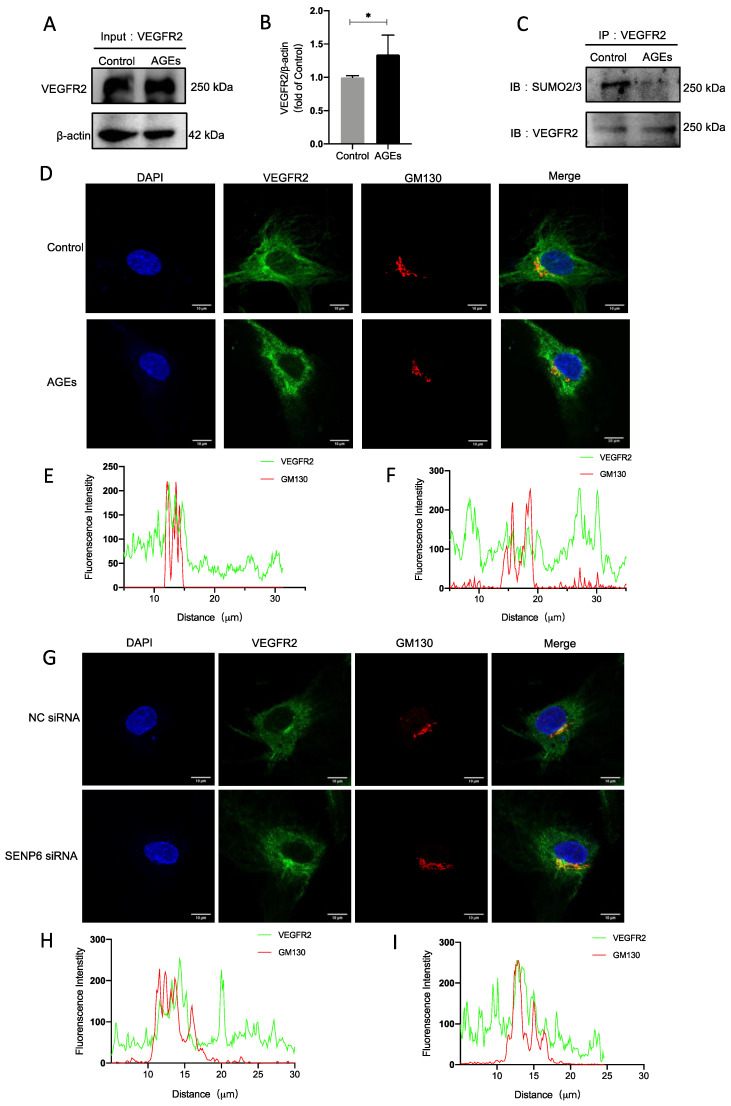
VEGFR2 SUMOylation retains VEGFR2 in the Golgi. (**A**) The total protein indicated in VEGFR2 cell lysates was determined by Western blot. (**B**) Analysis of protein expression level results, *n* = 7, * *p* < 0.05 vs. control. (**C**) The effects of AGEs on the SUMOylation of VEGFR2 were detected by Co-IP followed by Western blot analysis, *n* = 5. (**D**) Immunofluorescence staining, VEGFR2 (green), GM130 (red), nucleus (blue), *n* = 3, Scale bar: 10 μm. (**E**,**F**) The green and red fluorescence density and localized distance were measured, (**E**): control, (**F**): AGEs. (**G**) Immunofluorescence staining, VEGFR2 (green), GM130 (red), nucleus (blue), *n* = 3, Scale bar: 10 μm. (**H**,**I**) The green and red fluorescence density and localized distance were measured, (**H**): NC siRNA, (**I**): SENP6 siRNA. * indicates significance between the indicated groups.

**Figure 5 ijms-24-02544-f005:**
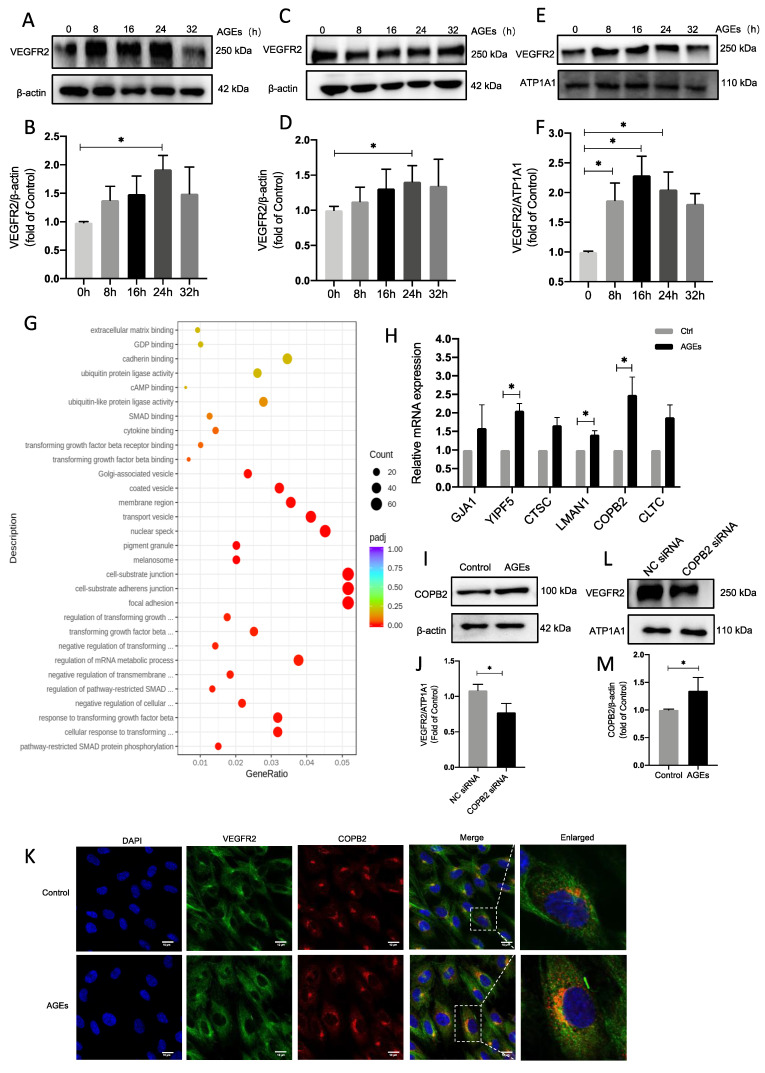
COPB2 mediates enhanced transport of VEGFR2 to membranes. (**A**,**B**) AGEs induce total VEGFR2 protein expression in endothelial cells. 100 µg/mL AGEs were applied to HUVECs stimulated for 0, 8, 16, 24, and 32 h, *n* = 6, * *p* < 0.05 vs. 0 h. (**C**–**F**) Endothelial cell cytoplasm cytosol was isolated and extracted with the kit. 100 µg/mL AGEs were applied to HUVECs stimulated for 0, 8, 16, 24, and 32 h VEGFR2 protein expression for cytoplasmic protein (**C**), cytosolic protein (**E**), *n* = 6, * *p* < 0.05 vs. 0 h. (**G**) Bubble plot GO enriched to upregulate the expression of proteins associated with Golgi transport. (**H**) qPCR detection of mRNA levels of the first few vesicles transport-related proteins, *n* = 4, * *p* < 0.05 vs. 0 h. (**I**,**J**) AGEs stimulation followed by detection of COPB2 protein expression levels in VEGFR2 expression, *n* = 7, * *p* < 0.05 vs. 0 h. (**K**,**L**) Immunofluorescence detection of COPB2 fluorescence intensity, DAPI (blue), VEGFR2 (green), and COPB2 (red). (**M**) Transfection of COPB2 siRNA to detect cell membrane VEGFR2 expression levels. * indicates significance between the indicated groups.

**Figure 6 ijms-24-02544-f006:**
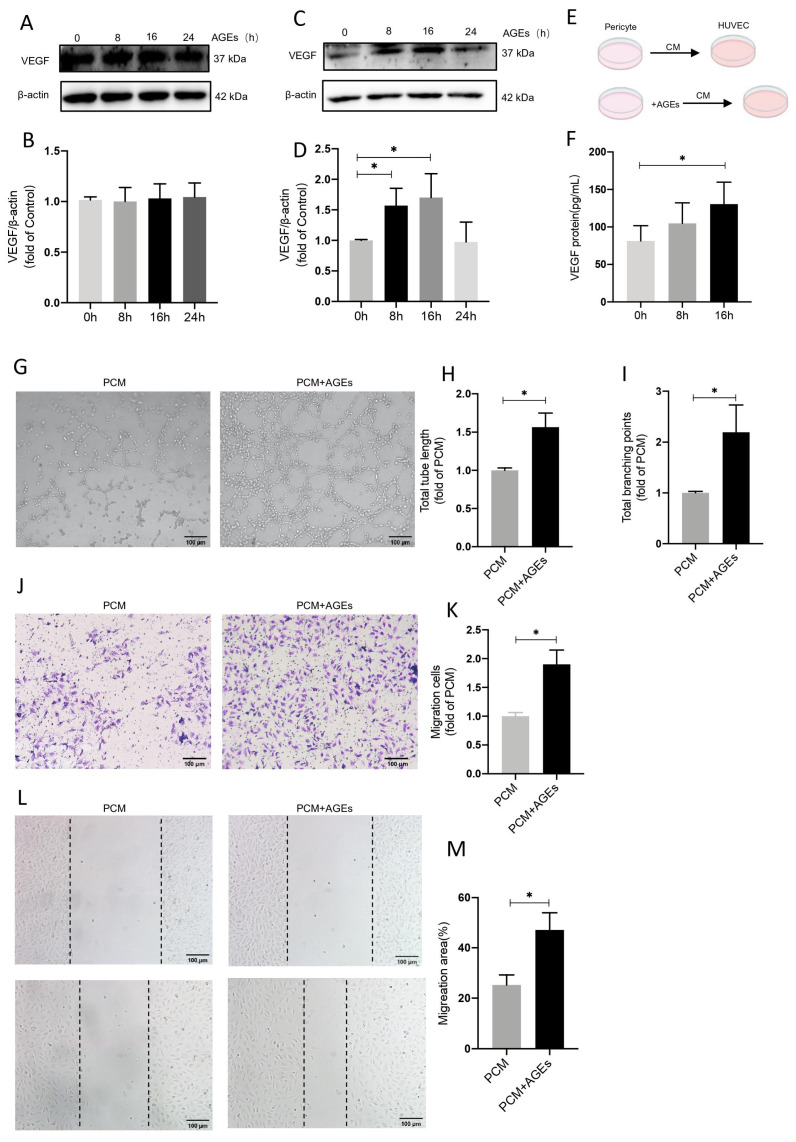
AGEs stimulate increased secretion of VEGF by pericytes and promote endothelial cell angiogenesis. (**A**,**B**) The temporal effect of AGEs stimulation on HUVECs was studied by Western blot, showing VEGF protein expression at 0, 8, 16, and 24 h, *n* = 6. (**C**,**D**) Temporal effect of AGEs stimulation on pericytes studied by Western blot, showing VEGF protein expression at 0, 8, 16, and 24 h of expression, *n* = 6, * *p* < 0.05 vs. 0 h. (**E**) Supernatant from pericytes was collected as a conditioned medium (CM) for endothelial cell culture. (**F**) Elisa assay for VEGF expression in the supernatant of pericyte medium. (**G**–**I**) Formation of HUVECs tubes in Matrigel assay after stimulation of endothelial cells with CM with or without AGEs, respectively, *n* = 4, * *p* < 0.05 vs. PCM, Scale bar: 100 μm. (**J**,**K**) Migration capacity in transwell migration assay, *n* = 4, * *p* < 0.05 vs. PCM, Scale bar: 100 μm. (**L**,**M**) Migration of HUVECs at specified times as determined by scratch tests. *n* = 4, * *p* < 0.05 vs. PCM, Scale bar: 100 μm. * indicates significance between the indicated groups.

**Table 1 ijms-24-02544-t001:** The primer sequences in this study (5′→ 3′).

Name	Sequence
SENP6 forward	CGGGTGCGGCCATTT
SENP6 reverse	GCCGTGGGTTCCCAAGA
Gja1 forward	GACTGCGGATCTCCAAAATA
Gja1 reverse	CTGTAATTCGCCCAGTTTTG
YIPF5 forward	GTAGCAGATGGCAGCATCAT
YIPF5 reverse	TGCCAGCCAGTAGCAATGTG
CTSC forward	CCCTGGGAGATATGATTAGGAGA
CTSC reverse	CAGTCAGTGGTGCAGGTTTG
LMAN1 forward	GGTGATCTTTCCATTGGTG
LMAN1 reverse	TCCGCTGTATGGTTACTTTG
COPB2 forward	CTTCCTGTTCGAGCTGCAAAG
COPB2 reverse	CACTCTAATCTGCATGTCATCCG
CLTC forward	GCACTGAAAGCTGGGAAAACT
CLTC reverse	CTGCAAGGCTAGAATGGCGA

**Table 2 ijms-24-02544-t002:** Sequences of siRNA in this study (5′→ 3′).

Name	Sequence
NC siRNA Sense	UUCUCCGAACGUGUCACGUTT
NCsiRNAAntisense	ACGUGACACGUUCGGAGAAT
Nrf2siRNA Sense	GGAGGCAAGAUAUAGAUCUTT
Nrf2siRNAAntisense	AGAUCUAUAUCUUGCCUCCT
SENP6siRNA Sense	CCAAGGUGUUGAACGUAUATT
SENP6siRNAAntisense	UAUACGUUCAACACCUUGGTT
COPB2siRNA Sense	CCCAUUAUGUUAUGCAGAUTT
COPB2siRNAAntisense	AUCUGCAUAACAUAAUGGGTT

## Data Availability

The data presented in this study are available on request from the corresponding author.

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
