# Peer review of "SENP6-Mediated deSUMOylation of VEGFR2 Enhances Its Cell Membrane Transport in Angiogenesis"

_ijms, 2023, doi:10.3390/ijms24032544_

Round 1

Reviewer 1 Report

Abstract:

The authors may introduce more the current knowledge of the role of SUMO in post-translational modification and its relationship to AGE.

Introduction

p.2 line 59: “RAGE” is not defined previously

Results

Fig 3B-D: why did SENP6 decrease after 12h of AGE treatment?

Discussion

Is the AGE-mediated Nurf2-SENP6-VEGFR reaction different from any other oxidative stress? What is the uniqueness?

Author Response

Dear Editors and reviewers:

Re: Manuscript ID: ijms-2147197 and Title: SENP6-mediated deSUMOylation of VEGFR2 enhances its cell membrane transport in angiogenesis

Thank you for your precious comments and advice. Those comments are all valuable and very helpful for revising and improving our paper, as well as the important guiding significance to our research. We have studied the comments carefully and have made correction which we hope meet with approval. This letter is our point-by-point response to the comments raised by the reviewers. The main corrections in the paper and the responses to the reviewer’s comments are as flowing.

We would like to thank you for allowing us to resubmit a revised manuscript and we highly appreciate your time and consideration. If there are any other modifications we could make, we would like very much to modify them and we really appreciate your help.

Sincerely,

Xiaohua Guo

Response to the reviewer's comments:

#1: The authors may introduce more the current knowledge of the role of SUMO in post-translational modification and its relationship to AGE.

 Response: We are very grateful for your comments on the manuscript. We currently know very little literature on AGEs and SUMOylation. It has been shown that AGEs induce endogenous ERK5 SUMOylation and thereby mediate endothelial dysfunction[1]. Indeed, in this study, the SUMO-mediated modification of AGEs and related proteins is one of the novelties of this study.

#2: p.2 line 59: “RAGE” is not defined previously

 Response: We sincerely thank you for your careful reading and apologize for not defining this earlier in the original draft. We agree with the comment and we have corrected the “RAGE” into “the receptor for advanced glycation end products” (p.2 line 59).

#3: Fig 3B-D: why did SENP6 decrease after 12h of AGE treatment?

 Response:  We deeply appreciate your suggestion. As far as we can explain, the stable expression of different proteins varies, and it is possible that some sort of surrogate mechanism is involved. That is, once the protein concentration reaches a certain level, it returns to physiological levels when its action is complete.

#4: Is the AGE-mediated Nurf2-SENP6-VEGFR reaction different from any other oxidative stress? What is the uniqueness?

Response: We feel great thanks for your professional review work on our article. As far as we know from the literature, AGEs can activate NF-κB to increase the expression of inducible NO synthase (iNOS) through RAGE/RhoA/ROCK-mediated and AMPK-mediated signaling pathways, which subsequently promote the generation of NO in endothelial cells [2]. AGEs were also reported to increase the expression and activity of NADPH oxidase in endothelial cells, which is an important source of oxidative stress in diabetic cardiovascular complications [3, 4]. However, in the current study, we did not focus on AGEs-mediated oxidative stress and did not conduct relevant experiments to detect oxidative stress levels. We concentrated on the Nrf2-SENP6-VEGFR2 signaling pathway mediated by AGEs in angiogenesis. The role of this signaling pathway in oxidative stress also provides us with good ideas for future experiments, and we will explore and implement relevant experiments on the role of this signaling pathway in oxidative stress in the future.

  1. Woo, C. H., T. Shishido, C. McClain, J. H. Lim, J. D. Li, J. Yang, C. Yan and J. Abe. "Extracellular signal-regulated kinase 5 sumoylation antagonizes shear stress-induced antiinflammatory response and endothelial nitric oxide synthase expression in endothelial cells." Circ Res 102 (2008): 538-45. 10.1161/circresaha.107.156877.
  2. Tang, S. T., Q. Zhang, H. Q. Tang, C. J. Wang, H. Su, Q. Zhou, W. Wei, H. Q. Zhu and Y. Wang. "Effects of glucagon-like peptide-1 on advanced glycation endproduct-induced aortic endothelial dysfunction in streptozotocin-induced diabetic rats: Possible roles of rho kinase- and amp kinase-mediated nuclear factor κb signaling pathways." Endocrine 53 (2016): 107-16. 10.1007/s12020-015-0852-y.
  3. Ren, X., L. Ren, Q. Wei, H. Shao, L. Chen and N. Liu. "Advanced glycation end-products decreases expression of endothelial nitric oxide synthase through oxidative stress in human coronary artery endothelial cells." Cardiovasc Diabetol 16 (2017): 52. 10.1186/s12933-017-0531-9.
  4. Chen, Y. H., Z. W. Chen, H. M. Li, X. F. Yan and B. Feng. "Age/rage-induced emp release via the nox-derived ros pathway." J Diabetes Res 2018 (2018): 6823058. 10.1155/2018/6823058.

Reviewer 2 Report

In this paperijms-2147197-peer-review-v1, the authors found that SENP6-mediated deSUMOylation of VEGFR2 enhanced its cell membrane transport during angiogenesis. The structure of the paper is reasonable and the content is comprehensive. However, there are still the following problems and suggestions for further revision and improvement:

1. It is suggested to increase the cytotoxicity test of AGEs on HUVECs.

2. It is suggested to mark the meaning of CM and CN in Figure 1A.

3. In Figure 1, it is suggested that the time setting of protein expression level of Nrf2 should be consistent with the expression level of VEGFR2.

4. In Figure 4, the protein level of VEGFR2 in AGEs group does not seem to increase significantly based on immunofluorescence analysis.

5. In Figure 4, it is recommended to add fluorescence intensity statistical graph to Figure G.

6. It is recommended to place pictures with same magnification and the same statistical chart in Transwell and Wound healing charts.

7. The picture of the tube formation is not clear and the magnification is inconsistent.

8. In the discussion section, it is mentioned that VEGFR2 may function as a downstream gene of Nrf2, and it is recommended to supplement relevant experiments to prove the conclusion.

9. Please unify the format of fonts and symbols in full-text figures.

10. It has been reported in the literature that AGEs inhibited HUVECs migration (Li et al., Int J Mol Sci. 2017, 18(2):436. doi: 10.3390/ijms18020436.). In the current study, AGEs was shown to accelerate the migration of HUVECs. The authors are suggested to discuss the contradictory findings.

Author Response

Dear Editors and Reviewers:

Re: Manuscript ID: ijms-2147197 and Title: SENP6-mediated deSUMOylation of VEGFR2 enhances its cell membrane transport in angiogenesis

Thank you for your precious comments and advice. Those comments are all valuable and very helpful for revising and improving our paper, as well as the important guiding significance to our research. We have studied the comments carefully and have made correction which we hope meet with approval. This letter is our point-by-point response to the comments raised by the reviewers. The main corrections in the paper and the responses to the reviewer’s comments are as flowing.

We would like to thank you for allowing us to resubmit a revised manuscript and we highly appreciate your time and consideration. If there are any other modifications we could make, we would like very much to modify them and we really appreciate your help.

Sincerely,

Xiaohua Guo

Response to the reviewer's comments:

#1: It is suggested to increase the cytotoxicity test of AGEs on HUVECs.

Response: We are grateful for the suggestion. As suggested by you, we have added Cell Counting Kit-8(CCK8) experiments to verify the cytotoxicity of AGEs on HUVECs. The results showed no cytotoxicity to HUVECs after AGEs stimulation and an increase in their activity, and this part of the results was placed in Supplemental Fig. S1, A. And we have added a description of this section in the Materials and Methods section (p.19, line 461-466). In addition, our previously published literature shows the same conclusion[1].

 cytotoxicity test of AGEs,n=3, *P < 0.05 vs. Control

“Cell Counting Kit-8 (CCK8) was used to measure cytotoxicity (Dojindo Molecular Technologies Inc., Kumamoto, Japan). Cells were seeded in 96-well culture plates and then treated in groups. After removing the media, CCK-8 (0.5 mg/mL) was added to each well for 4 hours. At 450 nm, the absorbance was measured. HUVEC proliferation was assessed directly using optical density (OD).”

#2: It is suggested to mark the meaning of CM and CN in Figure 1A.

Response: Thank you for your suggestion. We are sorry for our carelessness. We have redefined CN (cytonucleus) and CP (cytoplasm) in Figure 1A.

#3: In Figure 1, it is suggested that the time setting of protein expression level of Nrf2 should be consistent with the expression level of VEGFR2.

Response: We completely agree with this valuable suggestion by you. As a matter of fact, we attempted to carry out such an analysis before submitting the original manuscript. It eventually showed that the entry of Nrf2 into the nucleus was evident at 6h and that VEGFR2 expression started to increase at 8h expression, just not statistically different and relatively consistent in terms of time points. Whereas Nrf2 is upstream of VEGFR2, we then followed 0, 2, 4 and 6h to detect the time of Nrf2 entry into the nucleus. We followed 0, 8, 16, 24 and 32h to detect the time of VEGFR2 expression. The time relationship between the two was set up in an upstream-downstream relationship. As such we gave up the attempt to follow a consistent time point for detection.

#4:  In Figure 4, the protein level of VEGFR2 in AGEs group does not seem to increase significantly based on immunofluorescence analysis.

Response: Thank you for your advice. We have replaced the image in Figure 4D AGEs group and the immunofluorescence of VEGFR2 is increased.

#5: In Figure 4, it is recommended to add fluorescence intensity statistical graph to Figure G.

Response: Thank you for your suggestion. According to your comment, we did a fluorescence intensity statistical graph in Figures 4H-I. Consistent with this, we have illustrated accordingly in the figure legend to Figure 4, “The green and red fluorescence density and localized distance were measured, H: NC siRNA, I: SENP6 siRNA “(p.11 line 208-209).

#6:  It is recommended to place pictures with same magnification and the same statistical chart in Transwell and Wound healing charts.

Response: Thank you for your careful review. To be clearer and in accordance with your concerns, we have changed pictures with the same magnification and the same statistical chart in Transwell and Wound healing charts.

#7:  The picture of the tube formation is not clear and the magnification is inconsistent.

Response: We are grateful for your pointing out this problem. We have replaced the images with clearer pictures of the tube and have changed them to the same magnification.

#8:  In the discussion section, it is mentioned that VEGFR2 may function as a downstream gene of Nrf2, and it is recommended to supplement relevant experiments to prove the conclusion.

Response: We deeply appreciate your suggestion. In Figure 1F-I, we demonstrated with Nrf2 inhibitors (ML385) and Nrf2 siRNAs that VEGFR2 expression decreases following inhibition of Nrf2 or downregulation of Nrf2 expression, which could demonstrate that VEGFR2 may function as a downstream gene of Nrf2.

#9:   Please unify the format of fonts and symbols in full-text figures.

Response: Thanks for your careful checks. We are sorry for our carelessness. Based on your suggestion, we have made the corrections to unify the format of fonts and symbols in full-text figures.

#10:   It has been reported in the literature that AGEs inhibited HUVECs migration (Li et al., Int J Mol Sci. 2017, 18(2):436. doi: 10.3390/ijms18020436.). In the current study, AGEs was shown to accelerate the migration of HUVECs. The authors are suggested to discuss the contradictory findings.

Response: We feel very grateful for your professional suggestion and sincerely appreciate your valuable comments. We have added more references to discuss the findings in the revised manuscript. (p.15 line 275-285).

“For many years, scientists have been studying the effects of AGEs on endothelial cells. However, whether AGEs promote or inhibit HUVEC proliferation and migration is still being debated. Our previous study showed that AGEs (100ug/mL) promoted the proliferation, migration, and tube formation of HUVECs [1, 2]. Accordingly, several experimental studies have come to the same conclusions as ours [3-5]. However, there are also kinds of literature showing that AGEs inhibit, migration and tubule formation [6, 7]. There is also literature that systematically addresses this paradox effect of glycosylation on angiogenesis, as AGEs can play contradictory roles at different stages of angiogenesis[8]. Possible explanations include (1) different AGE concentrations in different studies, and (2) different AGE preparation methods in different studies. (3) Different culture conditions may result in the production of different active compounds in AGEs.”

  1. Wang, Q., A. Fan, Y. Yuan, L. Chen, X. Guo, X. Huang and Q. Huang. "Role of moesin in advanced glycation end products-induced angiogenesis of human umbilical vein endothelial cells." Sci Rep 6 (2016): 22749. 10.1038/srep22749.
  2. Li, P., D. Chen, Y. Cui, W. Zhang, J. Weng, L. Yu, L. Chen, Z. Chen, H. Su, S. Yu, et al. "Src plays an important role in age-induced endothelial cell proliferation, migration, and tubulogenesis." Front Physiol 9 (2018): 765. 10.3389/fphys.2018.00765.
  3. Tezuka, M., N. Koyama, N. Morisaki, Y. Saito, S. Yoshida, N. Araki and S. Horiuchi. "Angiogenic effects of advanced glycation end products of the maillard reaction on cultured human umbilical cord vein endothelial cells." Biochem Biophys Res Commun 193 (1993): 674-80. 10.1006/bbrc.1993.1677.
  4. Devi, M. S. and P. R. Sudhakaran. "Differential modulation of angiogenesis by advanced glycation end products." Exp Biol Med (Maywood) 236 (2011): 52-61. 10.1258/ebm.2010.010087.
  5. Xie, Y., S. J. You, Y. L. Zhang, Q. Han, Y. J. Cao, X. S. Xu, Y. P. Yang, J. Li and C. F. Liu. "Protective role of autophagy in age-induced early injury of human vascular endothelial cells." Mol Med Rep 4 (2011): 459-64. 10.3892/mmr.2011.460.
  6. Li, Y., Y. Chang, N. Ye, D. Dai, Y. Chen, N. Zhang, G. Sun and Y. Sun. "Advanced glycation end products inhibit the proliferation of human umbilical vein endothelial cells by inhibiting cathepsin d." Int J Mol Sci 18 (2017): 10.3390/ijms18020436.
  7. Zhao, Y., X. Wang, S. Yang, X. Song, N. Sun, C. Chen, Y. Zhang, D. Yao, J. Huang, J. Wang, et al. "Kanglexin accelerates diabetic wound healing by promoting angiogenesis via fgfr1/erk signaling." Biomed Pharmacother 132 (2020): 110933. 10.1016/j.biopha.2020.110933.
  8. Roca, F., N. Grossin, P. Chassagne, F. Puisieux and E. Boulanger. "Glycation: The angiogenic paradox in aging and age-related disorders and diseases." Ageing Res Rev 15 (2014): 146-60. 10.1016/j.arr.2014.03.009.

Reviewer 3 Report

1、Should the sixth part of the experimental results " AGEs stimulate increased secretion of VEGF by pericytes and promote endothelial cell angiogenesis " be introduced in the first half of the results as the relationship between AGE and angiogenesis? The position order of the research results should conform to the logic of the experimental idea.

2、The fifth part of the experimental results only studies the relationship between COPB2 and AGEs, but is there a direct or brief relationship between the core ubiquitination modification of the SENP family and COPB2? It is worth further clarification.

3、It is suggested that the author increase further results of in vivo study.

4、In the part of original images for Blots, the markers for specific molecular size are missing. It is suggested that the author should provide the nearby markers for destination strip.

Author Response

Dear Editors and Reviewers:

Re: Manuscript ID: ijms-2147197 and Title: SENP6-mediated deSUMOylation of VEGFR2 enhances its cell membrane transport in angiogenesis

Thank you for your precious comments and advice. Those comments are all valuable and very helpful for revising and improving our paper, as well as the important guiding significance to our research. We have studied the comments carefully and have made correction which we hope meet with approval. This letter is our point-by-point response to the comments raised by the reviewers. The main corrections in the paper and the responses to the reviewer’s comments are as flowing.

We would like to thank you for allowing us to resubmit a revised manuscript and we highly appreciate your time and consideration. If there are any other modifications we could make, we would like very much to modify them and we really appreciate your help.

Sincerely,

Xiaohua Guo

Response to the reviewer's comments:

#1: Should the sixth part of the experimental results " AGEs stimulate increased secretion of VEGF by pericytes and promote endothelial cell angiogenesis " be introduced in the first half of the results as the relationship between AGE and angiogenesis? The position order of the research results should conform to the logic of the experimental idea.

Response: Thank you very much for your comments and suggestions. As a matter of fact, we attempted to carry out such an order before submitting the original manuscript. However, this study follows the internal logic of protein “expression”- “modification”- “transport”. Firstly, AGEs upregulate VEGFR2 expression by inducing an increase in Nrf2 entry into the nucleus. Secondly, VEGFR2 is deSUMOylated by SENP6, which in turn reduces retention in Golgi, and then increases translocation to the cell membrane via COPB2. Ultimately, VEGFR2 translocated to the membrane binds to VEGF secreted by pericytes to trigger the downstream signaling pathway leading to angiogenesis. Since the VEGF-VEGFR2 signaling pathway has been well studied and is well understood, we do not dwell on it here. In summary, this study ultimately chose such a logical sequence to enumerate the results.

#2: The fifth part of the experimental results only studies the relationship between COPB2 and AGEs, but is there a direct or brief relationship between the core ubiquitination modification of the SENP family and COPB2? It is worth further clarification.

Response: We are extremely grateful for your pointing out this problem. Actually, in the present experiments, we found an increase in COPB2 expression following AGEs stimulation, which was used to reflect the enhanced translocation of VEGFR2 to the cell membrane after post-translational modification. SENP6 was to de-SUMOlate VEGFR2. Here, they can be seen as two separate pathways, one for “modification” and the other for “translocation”. Therefore, it is not necessary to focus on the relationship between the two.

#3: It is suggested that the author increase further results of in vivo study.

Response: We understand that in vivo studies would be more convincing to further validate the effect of the Nrf2-SENP6-VEGFR2 signaling pathway on AGE-induced angiogenesis. The lack of in vivo experiments is a shortcoming of this article.

In the current study, we used the mouse aortic ring angiogenesis assay. The results showed that treatment with AGEs(B) significantly increased the number of germinating vessels compared to control(A). This is consistent with our previously published literature[1, 2].

This article has been able to illustrate what we are trying to express at the cellular level. In addition, in vivo experiments are underway (e.g., construction of the AGEs model, which will probably take several months) and this will necessarily be carried out in full in the future once it is feasible and will be continued in future work.        

#4: In the part of original images for Blots, the markers for specific molecular size are missing. It is suggested that the author should provide the nearby markers for destination strip.

Response: Thank you for your reminder. We have re-labeled the markers of the original bands and the molecular weight of the target and position.

  1. Li, P., D. Chen, Y. Cui, W. Zhang, J. Weng, L. Yu, L. Chen, Z. Chen, H. Su, S. Yu, et al. "Src plays an important role in age-induced endothelial cell proliferation, migration, and tubulogenesis." Front Physiol 9 (2018): 765. 10.3389/fphys.2018.00765.
  2. Chen, L., Y. Cui, B. Li, J. Weng, W. Wang, S. Zhang, X. Huang, X. Guo and Q. Huang. "Advanced glycation end products induce immature angiogenesis in in vivo and ex vivo mouse models." Am J Physiol Heart Circ Physiol 318 (2020): H519-h33. 10.1152/ajpheart.00473.2019.

Round 2

Reviewer 3 Report

Thank you for responding to the comments. I have no further concerns.